# The Fluorinated NAD Precursors Enhance FK866 Cytotoxicity by Activating SARM1 in Glioblastoma Cells

**DOI:** 10.3390/biology13090649

**Published:** 2024-08-23

**Authors:** Wei Ming He, Jian Yuan Yang, Zhi Ying Zhao, Weimin Xiao, Wan Hua Li, Yong Juan Zhao

**Affiliations:** 1State Key Laboratory of Chemical Oncogenomics, Peking University Shenzhen Graduate School, Shenzhen 518055, China; 2001112061@pku.edu.cn (W.M.H.); zzypku2020@163.com (Z.Y.Z.); 2Ciechanover Institute of Precision and Regenerative Medicine, School of Medicine, The Chinese University of Hong Kong, Shenzhen 518172, China; 221059039@link.cuhk.edu.cn; 3Shenzhen Academy of Metrology and Quality Inspection, Shenzhen 518110, China; xiaowm@smq.com.cn

**Keywords:** F-NR, CZ-48, NRKs, SARM1, NAMPT, FK866, NAD, cADPR

## Abstract

**Simple Summary:**

In tackling the daunting challenge of glioblastoma, a severe form of brain tumor, researchers have been exploring ways to disrupt its abnormal energy production, focusing on a molecule called NAD. The drug FK866, known to deplete NAD, shows potential in curbing tumor growth but faces limitations when used alone. This study introduces a novel approach using fluorinated versions of NAD precursors, specifically a compound named F-NR, which when, combined with FK866, significantly boosts its effectiveness against glioblastoma cells. F-NR works by competing with the endogenous metabolism of NR, leading to reduced NAD levels and enhancing FK866’s ability to kill cancer cells. Another key finding is the role of SARM1, activated by one of F-NR’s metabolites, which contributes to the enhanced cell-killing effect. The sequence of events—NAD depletion followed by energy loss and ultimately, widespread cell death—highlights the potential of this strategy to improve FK866’s therapeutic impact. This research not only deepens our understanding of NAD metabolism in glioblastoma but also provides a potential strategy for treating the brain cancer.

**Abstract:**

Glioblastoma, a formidable brain tumor characterized by dysregulated NAD metabolism, poses a significant therapeutic challenge. The NAMPT inhibitor FK866, which induces NAD depletion, has shown promise in controlling tumor proliferation and modifying the tumor microenvironment. However, the clinical efficacy of FK866 as a single drug therapy for glioma is limited. In this study, we aim to disrupt NAD metabolism using fluorinated NAD precursors and explore their synergistic effect with FK866 in inducing cytotoxicity in glioblastoma cells. The synthesized analogue of nicotinamide riboside (NR), ara-F nicotinamide riboside (F-NR), inhibits nicotinamide ribose kinase (NRK) activity in vitro, reduces cellular NAD levels, and enhances FK866’s cytotoxicity in U251 glioblastoma cells, indicating a collaborative impact on cell death. Metabolic analyses reveal that F-NR undergoes conversion to fluorinated nicotinamide mononucleotide (F-NMN) and other metabolites, highlighting the intact NAD metabolic pathway in glioma cells. The activation of SARM1 by F-NMN, a potent NAD-consuming enzyme, is supported by the synergistic effect of CZ-48, a cell-permeable SARM1 activator. Temporal analysis underscores the sequential nature of events, establishing NAD depletion as a precursor to ATP depletion and eventual massive cell death. This study not only elucidates the molecular intricacies of glioblastoma cell death but also proposes a promising strategy to enhance FK866 efficacy through fluorinated NAD precursors, offering potential avenues for innovative therapeutic interventions in the challenging landscape of glioblastoma treatment.

## 1. Introduction

Glioblastoma, characterized by its aggressive behavior and therapeutic resistance, presents daunting challenges in both research and clinical management [1]. The infiltrative nature of glioblastoma, coupled with its limited responsiveness to standard treatment modalities such as surgical resection, chemotherapy, and radiation therapy, contributes to the persistently poor prognosis for affected patients. Consequently, there is a pressing need for innovative therapeutic approaches to ameliorate treatment outcomes and enhance patient survival in the context of glioblastoma.

The dysregulation of nicotinamide adenine dinucleotide (NAD) metabolism has emerged as a critical component in cancer pathogenesis, particularly in sustaining the rapid proliferation and survival of cancer cells. Elevated NAD levels have been associated with the enhancement of glycolysis and the facilitation of energy production pathways, thereby contributing to robust cancer cell growth and proliferation [2,3,4]. Notably, the amplification of nicotinamide phosphoribosyltransferase (NAMPT), the rate-limiting enzyme in NAD synthesis, underscores its pivotal role in sustaining NAD-dependent metabolic pathways essential for cancer cell survival. Thus, its small-molecule inhibitors, such as FK866, have emerged as a promising therapeutic agent against various cancers [5].

In the context of glioblastoma, therapeutic strategies targeting NAD metabolism have garnered attention. It was reported that NAMPT inhibitor FK866 induced cell death in C6 glioblastoma [6] and IDH1 mutant gliomas [7] by depleting NAD. It also demonstrated the synergistic effect in the combinatory use with Temozolomide, a chemotherapy drug via activating the ROS/JNK signaling pathway [8]. Another inhibitor, GMX1778, enhanced the anti-tumor efficacy of radiolabeled somatostatin analog ^177^Lu-DOTATATE treatment [9]. FK866 also decreased the glioblastoma stem-like cells (GSCs) self-renewal capacity, inhibited the in vivo GSC tumorigenicity, and decreased the GSC radioresistance [4].

Expanding research endeavors have delved into additional targets within the NAD metabolism network, including nicotinamide riboside kinases (NRKs) and sterile alpha and TIR motif containing 1 (SARM1) [10]. NRKs, involved in an alternative salvage pathway of NAD synthesis, catalyze the formation of NMN from nicotinamide riboside (NR), representing a potential target for modulating NAD metabolism [11]. Meanwhile, SARM1, a regulatable NAD-consuming enzyme, has exhibited significant implications in glioma progression and cellular death regulation [12]. Our lab has previously discovered a cell-permeant activator of SARM1, CZ-48, which induces non-apoptotic cell death in the cells overexpressing SARM1 [13]. SARM1 also suppresses glioma progression by inhibiting the proliferation of glioma cells and regulating microglial polarization [14].

Building upon these insights, the current study proposes a unique therapeutic approach involving the disruption of the NAD metabolism using a synthesized analogue of NR, ara-F nicotinamide riboside (F-NR). Strategically, F-NR acts as a competitive inhibitor of NRKs, consequently inhibiting the NRK pathway of NAD synthesis; concurrently, the resulting ara-F-NMN activates SARM1 and enhances NAD consumption. This approach aims to enhance the efficacy of NAMPT’s inhibitor, FK866, in depleting NAD levels, inducing ATP depletion, and ultimately promoting glioblastoma cell death.

## 2. Materials and Methods

### 2.1. Reagents

Lipofectamine 2000, Dulbecco’s Modified Eagle Medium, trypsin/TE solution, and penicillin/streptomycin solution were obtained from Thermo Fisher Scientific (Waltham, MA, USA). Fetal bovine serum was purchased from PAN Biotech (Aidenbach, Germany). Diaphorase, alcohol dehydrogenase, resazurin, FMN, perchloric acid, trifluoroacetic acid, NMN, NAD, ATP, and nicotinamide were obtained from Sigma-Aldrich (Saint Louis, MO, USA). CZ-48 and F-NMN were prepared in house [13]. Other chemicals were purchased from Sangon Biotech (Shanghai, China). NADase was extracted from *Neurospora crassa* [15]. The recombinant NMNAT1 [16] and ADP-ribosyl cyclase [17] were expressed and purified as described previously.

### 2.2. Cell Culture

HEK-293T and U251 cells were cultured in Dulbecco’s Modified Eagle Medium supplemented with 10% fetal bovine serum and 1% penicillin/streptomycin solution, and maintained in a standard humidified tissue culture incubator at 37 °C with 5% CO_2_.

### 2.3. Synthesis and Characterization of F-NR

F-NR was synthesized using the same procedure as CZ-48 [13], as illustrated in Figure 1A. Initially, **compound 1** was dissolved in CH_2_Cl_2_ and gradually added to a 33%HBr/CH_3_COOH solution under cold conditions. The reaction proceeded at room temperature overnight with stirring, and the product was purified via column chromatography. Subsequently, **compound 2** was dissolved in anhydrous acetonitrile, followed by the addition of nicotinamide. The mixture was stirred at room temperature for 24 h, and purification through column chromatography yielded **compound 3**. This product was then dissolved in methanol, and anhydrous potassium carbonate was introduced. Upon completion of the reaction, neutralization to pH 7 was achieved by adding diluted hydrochloric acid to eliminate the protective group. Following purification, F-NR was successfully obtained and then characterized by NMR and mass spectrometry.

### 2.4. Expression and Purification of NRKs

The recombinant proteins, NRK1 and NRK2, were prepared as follows. The CDS region of human NRK1 (NM_017881.3, NCBI) and NRK2 (NM_170678.2, NCBI) genes were subcloned in the prokaryotic expression vector pET22b, which were then transformed to the bacterial strain Rosetta (DE3). Following IPTG-induced expression, the proteins were purified using a three-step process involving nickel column affinity chromatography, anion exchange, and size exclusion chromatography. The purity of the proteins was verified by SDS-PAGE.

### 2.5. Measurement of the In Vitro Enzymatic Activity of NRKs

Measurement of the in vitro enzymatic activity of NRKs was based on the phosphorylation reaction converting NR to NMN. Briefly, the reaction mixture consisted of 20 mM Hepes pH 7.2, 5 mM DTT, 1 mM ATP, 5 mM MgCl_2_, and 500 µM NR in a final volume of 200 µL. The reaction was initiated by the addition of 1 µg of the recombinant NRK enzyme to the reaction mixture. At various time points, the enzyme was removed by filtration using a MultiScreen^®^ 96-well Ultrafiltration Plate with Ultracel^®^-10 membrane kD (Merck Millipore, St. Louis, MO, USA), and the nucleotides were analyzed by HPLC equipped with an AG MP-1 anion exchange column (Bio-Rad, Hercules, CA, USA) and eluted with a trifluoroacetic acid gradient. The rate of NMN production was monitored by cycling assay. The inhibition of NRKs’ activity was calculated by the rate of NMN production when F-NR was included in the reaction.

### 2.6. Nucleotides Extraction and Measurement

The nucleotides were extracted, and the levels of NAD, cADPR, NMN, and ATP were measured as described previously [13]. Briefly, the cells were lysed with perchloric acid, and following centrifugation, the NAD content in the supernatant was determined using a cycling assay [18]. The amounts of cADPR and NMN were also measured by cycling assay following conversion to NAD via the reactions catalyzed by ADP-ribosyl cyclase [18] and NMNAT1 [19], respectively. The ATP level was determined by a luciferase assay [13,20]. The resulting protein pellets were re-dissolved in 1 M NaOH and quantified using the Bradford assay (Quick Start™ Bradford Kit, Bio-Rad, Hercules, CA, USA). The results are presented as picomole NAD (or cADPR, NMN, ATP) per milligram of total proteins.

### 2.7. Data Acquisition for UPLC-HRMS-Based Metabolomics Analysis

Cells were treated with 100 µM F-NR for 48 h, untreated as controls, and lysed with methanol. After centrifugation, the 12 samples were analyzed using an untargeted UPLC-HRMS approach. A high-resolution tandem mass spectrometer Orbitrap Exploris 120 (Thermo Fisher, Waltham, MA, USA) was coupled to a Vanquish Flex UPLC system (Thermo Fisher, Waltham, MA, USA) equipped with an ACQUITY BEH Amide column (100 mm × 2.1 mm, 1.7 µm, Waters). The column oven was maintained at 60 °C. Mobile phase A was a water solution containing 10 mM ammonium acetate and 10 mM ammonia, and phase B was acetonitrile. A 10-min gradient elution profile was applied with: 0–0.5 min, 1% B; 0.5–6.5 min, 1% to 99% B; 6.5–8.0 min, 99% B; 8.0–8.5 min, 99% to 1% B; 8.5–10.0 min, 1% B. The flow rate was 0.35 mL/min and the injection volume for each sample was 4 µL. Data were acquired in both positive and negative ion modes, in which the ion spray voltage was +3800 V and −3400 V, respectively. The mass spectrometer was operated in the data-dependent acquisition (DDA) mode. Both MS1 and MS2 data were acquired in the profile mode. In each cycle, the primary acquisition range was 70–1050 Daltons. The first four signal ions with a signal accumulation intensity of more than 5000 were selected from the primary spectrum for secondary fragmentation scanning.

### 2.8. Data Analysis of Metabolomics Analysis

The acquired MS data pretreatments including peak picking, peak grouping, retention time correction, second peak grouping, and annotation of isotopes and adducts was performed using XCMS software [21]. LC-MS raw data files were converted into mzML format and then processed by the XCMS, MetaboAnnotation [22] toolbox implemented with the R software (v4.3.0). Each ion was identified by combining retention time (RT) and *m*/*z* data. The intensities of each peak were recorded and a three-dimensional matrix containing arbitrarily assigned peak indices (retention time–*m*/*z* pairs), sample names (observations) and ion intensity information (variables) was generated. The online KEGG, HMDB database [23] was used to annotate the metabolites by matching the exact molecular mass data (*m*/*z*) of samples with those from database. We also used an in-house fragment spectrum library of metabolites to validate the metabolite identification.

Student’s *t*-test was conducted to detect differences in metabolite concentrations between two groups. The *p-*value was adjusted for multiple tests using an FDR (Benjamini—Hochberg). To visualize the relative content of differentially metabolized substances between the two groups, we utilized the R package pheatmap. For the differential analysis results, we employed the R package ggplot2 to create a volcano plot. The criteria for identifying upregulated and downregulated metabolites were as follows: metabolites with a ratio of ≥1.5, a *p*-value of ≤0.05, and a Variable Importance in Projection (VIP) score greater than 1 were considered upregulated and were plotted in red; conversely, metabolites with a ratio of ≤2/3, a *p*-value of ≤0.05, and a VIP score greater than 1 were considered downregulated and were plotted in blue.

For MS2 spectrum annotation, the acquisition software (Xcalibur 4.0.27, Thermo Fisher, Waltham, MA, USA) was applied to open the raw data and manually annotated MS2 spectra. The mass tolerance between the observed and the theoretical value is 10 ppm.

### 2.9. qRT-PCR

Total RNA was extracted using the E.Z.N.A.^®^ Total RNA Kit I (Omega Bio-tek, Norcross, GA, USA). Subsequently, 1 µg of RNA was reverse transcribed into cDNA utilizing the TransScript First-Strand cDNA Synthesis SuperMix (TransGen Biotech, Beijing, China). Quantitative PCR was carried out using the One-Step Qrt-PCR SuperMix (TransGen Biotech, Beijing, China) on a Bio-Rad CFX machine (Bio-Rad, Hercules, CA, USA) following the manufacturer’s guidelines. The relative expression levels were determined by normalization with *Actin*, a housekeeping gene, utilizing the ΔCt method. The following primers were used: *Ido* (5’-AAATCCACGATCATGTGAACCCA-3’; 5’-ACCCTTCATACACCAGACCG-3’), *Narpt* (5’-TCCCTGGGTGGCGTCTATAA-3’; 5’-ATGAGTGGAGACCCGTCAGA-3’), *Nadsyn1* (5’-CAGGCTCGAATACGGATGG-3’; 5’-CGCACTGGAGCAGTCGTA-3’), *Nampt* (5’-CAGGGCTGCTTTTAACTCTGGT-3’; 5’-GATTTTGGAGGGATCTCGCT-3’), *Nrk1* (5’-TCTCCGGGATACTTTGATGGC-3’; 5’-CATCCAGGTACACAACTTCCCAT-3’), *Nrk2* (5’-AGGATGACTTCTTCAAGCCCC-3’; 5’-ATGGCCTCCATGTCCAGAGA-3’), *Sarm1* (5’-CTGGACAAGTGCATGCAAGA-3’; 5’-GGTGGCCTCCTGGTATTCGT-3’) and *Actin* (5’-CCTGGCACCCAGCACAAT-3’; 5’-GGGCCGGACTCGTCATACT-3’).

### 2.10. Cell Viability Assay

The cell viability was measured by the Sulforhodamine B (SRB) method [24]. Briefly, U251 cells were treated with drugs and incubated for various durations until the onset of cell death. After incubation, the cells were fixed with 10% trichloroacetic acid (TCA) for 2 h at 4 °C. The supernatant was then discarded, and the cells were washed with distilled water. SRB dye was added to each well and incubated for 5 min at room temperature. The wells were then washed with 1% acetic acid to remove excess dye. The acetic acid was discarded, and cell images were captured under a Nickon microscope. Then, the bound SRB dye was solubilized with 10 mM Tris-base, and the absorbance was measured at 540 nm using a microplate reader (Infinite M200 PRO, Tecan, Mannedorf, Switzerland). Cell viability was expressed as a percentage of the absorbance relative to the untreated controls.

### 2.11. Construction of SARM1-KO U251 Cell Line

To construct the SARM1 knockout U251 cell line, sgRNA targeting the coding region for Glu642 in SARM1 (5′-ATTGTGACTGCTTTAAGCTGCGG-3′) was subcloned to LentiCRISPR-V2 (#52961, Addgene). Lentiviral particles were prepared by transfecting HEK-293T cells with the lentivector, pMD2.G and psPAX2, followed by cell infection and selection of U251 cells by puromycin as described previously [25]. The SARM1-KO cell line was validated by the CZ-48 insensitivity. Briefly, the cells were treated with 100 µM CZ-48 for 24 h and the lysate was subjected to cADPR measurement.

### 2.12. Data Analysis

All the experiments were repeated at least three times. Data were presented as means ± SD. For multi-group comparisons, one-way analysis of variance (ANOVA) was employed, while unpaired Student’s *t*-tests were used for two-group comparisons, as specified in the figure legends. All statistical tests were two-tailed, with a *p*-value threshold of less than 0.05 deemed significant. The significance levels are denoted as follows: ns, not significant; *, *p* < 0.05; **, *p* < 0.01; ***, *p* < 0.001; ****, *p* < 0.0001.

## 3. Results

### 3.1. The Fluorinated NR (F-NR) Competitively Inhibits the Activity of NRKs and Affects NAD Metabolism

We hypothesized that inhibiting NRKs might be beneficial in NAD depletion-based cancer treatment since the NRK-mediated salvage pathway may rescue FK866-treated cancer cells. To test this hypothesis, we first synthesized a ribose-fluorinated NR (F-NR) through a three-step reaction, as shown in Figure 1A. The only structural difference between NR and F-NR lies in the 2′-substitution of the ribose, with NR having an (R)-hydroxyl group and F-NR having an (S)-fluoro group (Figure 1B).

Next, we expressed and purified recombinant NRK1 and NRK2 from an *E. coli* expression system using NTA-column combined with ion-exchange and gel filtration chromatography. The purified proteins exhibited the expected molecular weights (Figure 1C) and were enzymatically active, as demonstrated by the conversion of most NR to NMN within 60 min in HPLC analysis (Figure 1D). The unit activities were 12.03 and 17.28 pmol NR/µg/h for NRK1 and NRK2, respectively. In the NRK-catalyzed reactions, the production of NMN gradually decreased with increasing concentrations of F-NR, and the IC_50_ values of F-NR were close to the concentration of the substrate NR (Figure 1E), consistent with a competitive inhibition mechanism.

To evaluate the cellular effects of F-NR, we employed HEK-293 cells stably overexpressing SARM1 at a relatively low level [13]. We selected a relatively high concentration of F-NR based on preliminary dose-response studies that showed robust effects on NMN and NAD synthesis without causing significant cytotoxicity at a short time point. Upon treatment with F-NR, the levels of NMN (Figure 1F) and NAD (Figure 1G) were significantly decreased, consistent with the NRK-inhibitory activity of F-NR. This result also indicates that F-NR is cell-permeable, likely through the action of ENT transporters [26]. Additionally, we observed a significant increase in cellular cADPR levels in F-NR-treated cells (Figure 1H), suggesting that SARM1 might be activated by the F-NR-derived F-NMN increase and the NRK inhibition-induced NAD decrease.

Collectively, these results demonstrate that F-NR is metabolizable and might perturb NAD metabolism in cells.

### 3.2. F-NR Can Be Metabolized and Affects NAD Metabolism in Glioblastoma U251 Cells

To examine the impact of F-NR on glioblastoma, we treated U251 cells and assessed the NAD and cADPR levels. We used a lower concentration of 100 µM F-NR to investigate the effects on NAD levels over a longer period (two days). This was done to assess the cumulative impact of F-NR on NAD metabolism and to ensure that the effects were not due to cytotoxicity. Consistent with our findings in HEK-293 cells, cADPR increased while NAD decreased (Figure 2A). To further investigate the metabolism of F-NR and its impact on endogenous cell metabolism, we conducted untargeted metabolomics studies on both control and F-NR-treated cells. The cells were lysed and analyzed by UPLC-HRMS. Approximately 150 metabolites exhibited more than a 1.5-fold change, with 123 upregulated and 27 downregulated (Figure 2B). Due to the poor annotation of some compounds, the functional clustering was not clear. Among these differential metabolites, some were related to NAD metabolism. For instance, nicotinamide, a degradation product of F-NR, increased in F-NR-treated cells, and some lipid-related metabolites, including palmitoylcarnitine (CAR 16:0), lysophosphatidic acids (LysoPA), and lysophosphatidylinositol (LysoPI), were also upregulated (Figure 2C).

Derivatives of F-NR, including F-NMN (likely metabolized by NRKs) and F-NAD (further metabolized by NMNATs), were identified by MS1 signals in F-NR-treated cells (Figure 2C). The identity of these compounds was further verified through MS2 chemical structure profiling (F-NMN, Figure 2D). The original MS1 intensity plot showed that F-NR concentrations were significantly higher than those of F-NMN and F-NAD, with the latter two in the million range, similar to endogenous NAD (Figure 2E). Notably, endogenous NMN was undetectable by MS, aligning with its swift conversion into NAD by NMNATs. This finding indicated that F-NMN was less effectively transformed into its fluorinated NAD equivalent (F-NAD) by NMNATs, resulting in an F-NMN buildup. As F-NMN is known to activate SARM1 [13], we concluded that F-NMN accumulation was the primary driver of F-NR-mediated SARM1 activation (Figure 1H and Figure 2A).

In summary, F-NR was metabolized into F-NMN, which further activated endogenous SARM1 and then consumed NAD. Apart from NAD metabolism and potentially some lipid metabolism pathways, the overall metabolism of the cells was not significantly affected.

### 3.3. F-NR Enhances the Cytotoxicity of FK866 in U251 Glioma Cells

To explore the potential application of F-NR in cancer therapy via the NAD-depletion strategy, we first quantified the expression of key enzymes involved in the three NAD synthesis pathways in U251 glioma cells using qRT-PCR. These pathways include the de novo pathway (IDO and NADSYN1), the Preiss–Handler pathway (NAPRT and NADSYN1), and the salvage pathway (NAMPT, NRK1, and NRK2). Our results indicated that NAMPT was expressed at the highest level, followed by moderate expression of IDO, NADSYN1, NRK1, and NRK2, while NAPRT was undetectable (Figure 3A). This suggests that U251 cells primarily rely on the salvage pathway for NAD synthesis. Notably, SARM1, the F-NR-activable NADase, was also expressed in these cells (Figure 3A). These findings align with previous reports that inhibiting NAMPT can effectively kill glioblastoma cells and suggest that F-NR might synergistically enhance cell death by inhibiting NRKs and activating SARM1.

Subsequently, we treated U251 cells with FK866 and F-NR for two days and measured cellular NAD levels. FK866 treatment alone reduced NAD levels by approximately 80% and the combination of FK866 and F-NR resulted in the lowest NAD levels observed (Figure 3B), indicating that F-NR accelerates NAD depletion. Additionally, we measured cADPR levels, an indicator of SARM1 activation. The results confirmed that F-NR activates SARM1, while FK866 treatment decreased cADPR levels (Figure 3C), likely due to the depletion of NAD, the substrate for SARM1.

We then assessed cell viability using the SRB staining method. In the absence of F-NR, FK866 exhibited an EC_50_ of approximately 40 nM in killing U251 cells. However, in the presence of increasing concentrations of F-NR, the survival curves for FK866 shifted downward, resulting in lower EC_50_ values of 30 nM and 15 nM for 20 µM and 50 µM F-NR, respectively (Figure 3D). Without FK866, U251 cells tolerated F-NR concentrations as high as 200 µM. However, the addition of FK866 caused the survival curves to drop, with EC_50_ values of 100 µM and 30 µM in the presence of 10 nM and 20 nM FK866, respectively (Figure 3E). Microscopic images further illustrated the combinatory effect of F-NR and FK866; treatment with 20 nM FK866 or 50 µM F-NR alone did not significantly reduce cell numbers, but their combination resulted in a marked reduction (Figure 3F).

In summary, these results confirm that F-NR and FK866 synergistically enhance the cytotoxicity against U251 glioblastoma cells.

### 3.4. SARM1 Activation Enhances FK866’s Cytotoxicity in U251 Cells

Given that F-NR activates the NAD-consuming enzyme SARM1 in cells, we hypothesized that SARM1 activation plays a crucial role in the synergistic NAD-depletion effect. To test this hypothesis, we employed two analogues of NMN, ribose-fluorinated NMN (F-NMN) and CZ-48, to activate SARM1. The synthesis of these two compounds was described in our previous publication [13] and the cellular transportation might be through the transporters such as Slc12a8 [27]. As expected, in the presence of 50 µM of these compounds, the EC_50_ of FK866 dropped by approximately seven-fold (Figure 4A), an even more significant reduction than that observed with F-NR.

Next, we compared the cell viability curves between wild-type and SARM1-depleted U251 cells. In the presence of 50 µM CZ-48, wild-type cells were much more sensitive to FK866 (Figure 4B, solid lines), with the EC_50_ decreasing from 28 nM to 6 nM. However, SARM1-knockout cells did not show any change in sensitivity to FK866, regardless of the presence or absence of CZ-48 (Figure 4B, dashed lines). SRB imaging further confirmed that SARM1 knockout protected the cells from the combinatory treatments of CZ-48 and FK866 (Figure 4C).

The significant protective effect observed with SARM1 depletion prompted us to measure the cellular levels of related metabolites. In response to FK866 treatment, wild-type cells exhibited depleted NAD and cADPR levels, while ATP levels were maintained at approximately half of the baseline (Figure 4D, 1st and 2nd columns). When CZ-48 was added, the NAD and ATP levels decreased slightly, while the cADPR levels significantly increased in wild-type cells due to SARM1 activation (Figure 4D, 3rd columns). In the combined FK866 and CZ-48 treatment, both NAD and ATP were depleted, and cADPR levels were higher than those observed with CZ-48 treatment alone (Figure 4D, 4th columns). In SARM1-knockout cells, cADPR levels remained at basal levels (Figure 4D, middle chart), suggesting that SARM1 is the major ADP-ribosyl cyclase in U251 cells. In the FK866 or FK866/CZ-48 treated groups, NAD levels were higher in SARM1-knockout cells compared to wild-type cells (Figure 4D, upper chart), indicating that SARM1 contributes to NAD-depletion even in FK866-treated groups. ATP levels were only slightly affected (Figure 4D, lower chart).

In summary, our results demonstrate that SARM1 could be a viable drug target for NAD-depletion strategies in glioblastoma therapy.

### 3.5. SARM1 Activation Accelerates FK866-Induced NAD Depletion, ATP Depletion and Cell Death of U251 Cells

The results presented in Figure 4D prompted us to investigate the kinetics of CZ-48’s effects on metabolism and cell viability. Following FK866 treatment, the NAD levels decreased significantly within one day and sustained; while the cADPR levels gradually depleted over two days (Figure 5A,B, black lines). Co-treatment with CZ-48 led to an accumulation of cADPR during the first 2.5 days, followed by a sudden drop at day 4, coinciding with cell death (Figure 5B, red lines). The NAD levels decreased slightly by day 2.5 and dropped to zero by day 4 (Figure 5A, red lines). The trend of ATP depletion in the first 2.5 days were much slower compared to NAD in the FK866-alone group. However, co-treatment with CZ-48 significantly accelerated ATP depletion, especially after two days of treatment, leading to complete depletion by day 4 (Figure 5C). This ATP depletion was reflected in the cell viability results; in the FK866-alone group, approximately half of the cells remained viable after four days, whereas co-treatment with CZ-48 resulted in the death of the majority of cells (Figure 5D).

Combing the results from Figure 5, we found that very low levels of NAD could sustain the energetic needs of the cells. SARM1 activation further depleted the residual NAD in FK866-treated cells, again suggesting that reversible activation of SARM1 might be a viable strategy for enhancing the efficacy of FK866 in glioblastoma therapy.

## 4. Discussion

Our study builds upon the growing body of evidence supporting the application of NAMPT inhibitors, particularly FK866, in targeting glioblastoma [4,6,7,8,9]. Despite the failure of FK866 in clinical trials for other cancers [28], ongoing research continues to explore its potential in combination with other therapies for glioblastoma. We have elucidated the mechanism by which F-NR competitively inhibits NAD synthesis via the NRK pathway while concurrently being converted to F-NMN, which, along with reduced NAD levels, activates SARM1 and leads to further NAD consumption. This mechanism enables F-NR to synergistically enhance FK866’s cytotoxicity in glioblastoma cells. The addition of SARM1 activators, F-NMN and CZ-48, also significantly boosts FK866’s efficacy. Kinetics studies reveal the collaboration between NAMPT inhibition and SARM1 activation in depleting cytosolic NAD, leading to ATP depletion and ultimately cell death. Our metabolomics data further support these findings. This mechanism was summarized in Figure 6.

Notably, F-NR provides a feasible alternative to CZ-48 for the activation of SARM1. Its obvious advantage is that it can be easily synthesized in a typical biochemistry lab setting. CZ-48, as an analog of NMN, has been shown to activate SARM1 in cell cultures, making it a valuable tool for research [13]. Yet, the synthesis of CZ-48 is complicated, particularly due to the intricate process of incorporating the phosphorothioate group in the final stage [29]. This current study revealed that F-NR, lacking the phosphate moiety, is still able to activate SARM1, thereby offering greater accessibility to researchers. We have demonstrated that F-NR is cell-permeable and can be enzymatically converted into F-NMN by NRKs within cells, leading to the activation of SARM1.

## 5. Conclusions

In this study, we have demonstrated the potential of fluorinated nicotinamide riboside (F-NR) as an effective enhancer of FK866 cytotoxicity in glioblastoma cells through a novel mechanism. F-NR competes with the NRK pathway to inhibit NAD synthesis and is subsequently converted to F-NMN. This conversion, probably along with reduced NAD levels, activates SARM1, leading to further NAD consumption and exacerbating FK866’s effects. Our findings reveal a synergistic interaction between NAMPT inhibition and SARM1 activation, which collectively deplete cytosolic NAD, induce ATP depletion, and culminate in cell death.

Furthermore, our metabolomics data corroborate the biochemical pathways involved and provide a comprehensive understanding of how F-NR and FK866 interact at a molecular level. The ability to synthesize F-NR more easily than CZ-48 in typical biochemistry labs makes it a practical and accessible tool for researchers.

These insights not only reinforce the potential of NAMPT inhibitors like FK866 in glioblastoma treatment but also introduce F-NR as a promising adjunct to enhance therapeutic efficacy.

## 6. Limitations

While our findings are encouraging, several limitations should be acknowledged. First, our findings are primarily based on in vitro experiments. The cellular environment of glioblastoma within the human body is more complex, and the results observed in cell cultures may not fully translate to clinical scenarios. Second, the physiological impacts of metabolized fluorinated NAD precursors remain largely unexplored. The fluorinated metabolites could interact with endogenous metabolic or signaling pathways in ways that are currently not well understood, highlighting a critical area that requires further investigation.

## Figures and Tables

**Figure 1 biology-13-00649-f001:**
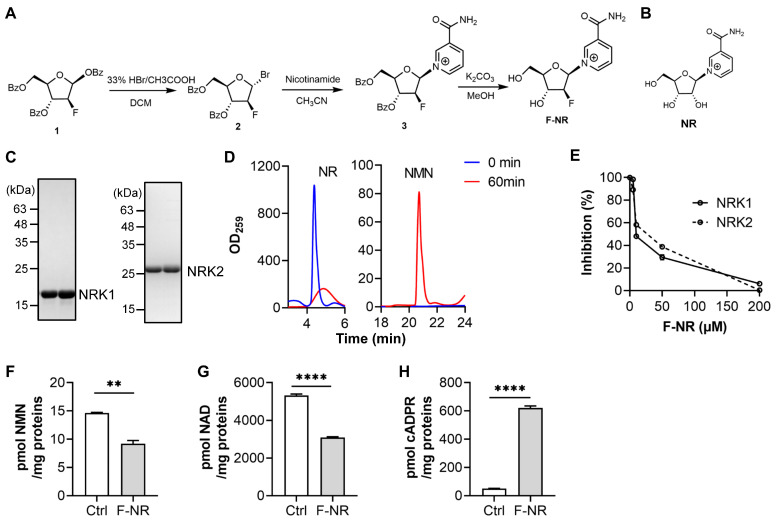
F-NR competitively inhibits NRKs and affects NAD metabolism. (**A**) The chemical synthesis route of F-NR. (**B**) Chemical structures of NR. (**C**) SDS-PAGE of purified recombinant NRK1 and NRK2. (**D**) HPLC detection of NRKs-catalyzed reactions. (**E**) Various concentrations of F-NR were added to reaction systems containing 50 µM NR and 20 ng NRKs, and incubated for 2.5 h. The produced NMN was quantified using the cycling assay [18]. (**F**–**H**) HEK-293 cells stably overexpressing a low level of SARM1 [13] were treated with 250 µM F-NR for 12 h, and intracellular levels of NMN (**F**), NAD (**G**), and cADPR (**H**) were measured using a cycling assay. All experiments were performed three times independently, and data are presented as mean ± standard deviation. Student’s *t*-test, **, *p* < 0.01; ****, *p* < 0.0001.

**Figure 2 biology-13-00649-f002:**
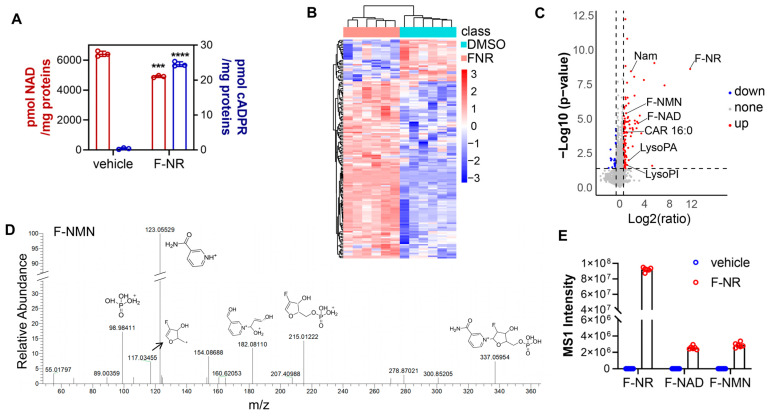
Metabolism of F-NR and its impact on other metabolites in U251 cells. U251 cells were treated with 100 µM F-NR for 2 days. (**A**) The cellular levels of cADPR and NAD were measured using the cycling assay. (**B**–**E**) Total metabolites were analyzed by untargeted metabolomics. The differential metabolites were analyzed by heat map (**B**). Metabolites were compared with a volcano plot of vehicle and F-NR-treated groups, where blue represents downregulation and red represents upregulation, and the dash line indicates a 1.5-fold difference (**C**). The metabolite F-NMN was analyzed by MS2 and the chemical structure of the fragmental ions were interpretated (**D**). F-NR and the derivatives including F-NMN and F-NAD were discovered and validated by MS2 analysis and the MS1 intensities were plotted (**E**). Data were presented as mean ± standard deviation. Student’s *t*-test, ***, *p* < 0.001; ****, *p* < 0.0001.

**Figure 3 biology-13-00649-f003:**
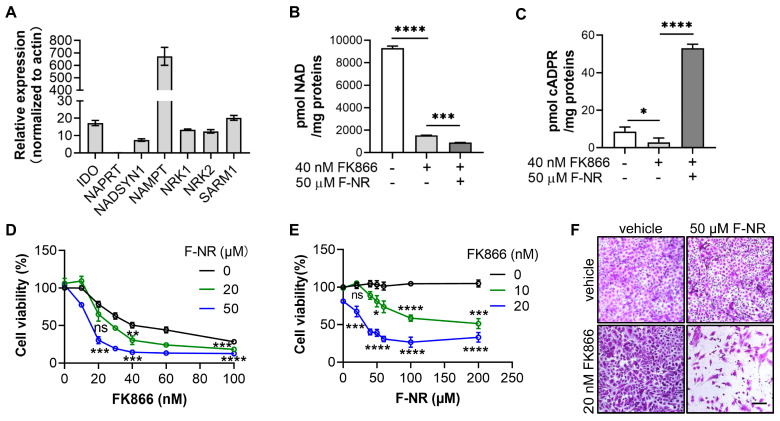
F-NR enhances the cytotoxicity of FK866 in U251 glioma cells. (**A**) mRNA levels of the key NAD metabolic enzymes in U251 cells were analyzed by qRT-PCR, normalized by the levels of the housekeeping gene, actin. (**B**,**C**) U251 cells were treated with the indicated drugs for two days. Intracellular levels of NAD (**B**) and cADPR (**C**) were measured using the cycling assay. (**D**,**E**) Viability of U251 cells was assessed after treatment with various concentrations of F-NR and FK866 for four days using an SRB staining method. (**D**) F-NR concentrations held constant, with increasing FK866 concentrations. (**E**) The converse, with FK866 concentrations held constant and increasing F-NR concentrations. (**F**) Microscopic imaging of U251 cells treated with 50 µM F-NR and 20 nM FK866 for four days, followed by SRB staining. Scale bar: 20 µm. All experiments were independently performed three times, and data are presented as mean ± standard deviation. One-way ANOVA test was used for statistical analysis. ns, not significant; *, *p* < 0.05; **, *p* < 0.01; ***, *p* < 0.001; ****, *p* < 0.0001.

**Figure 4 biology-13-00649-f004:**
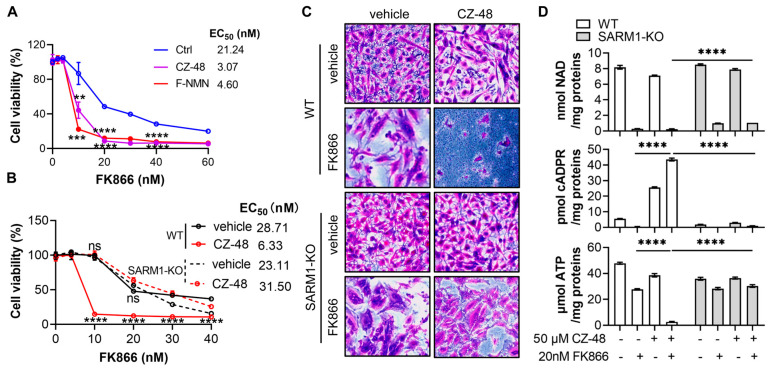
F-NMN and CZ-48 enhanced the cytotoxicity of FK866 on U251 glioma cells by activating SARM1. (**A**) U251 cells were treated with 50 µM F-NMN, or CZ-48, in the presence of different concentrations of FK866 for four days, and cell viability was measured using the SRB method. (**B**) Wild-type and SARM1-KO U251 cells were treated with 50 µM CZ-48 and different concentrations of FK866 for three days, and cell viability was measured using the SRB method. (**C**) Microscopic imaging of U251 cells treated with 50 µM CZ-48 and 20 nM FK866 for three days, followed by SRB staining. Scale bar: 10 µm. (**D**) U251 cells were treated with 20 nM FK866 and 50 µM CZ-48 alone or in combination for three days. The intracellular levels of NAD, cADPR, and ATP were measured. All experiments were independently performed three times, and data are presented as mean ± standard deviation. The statistical analysis among three samples in A was performed using a one-way ANOVA test, while comparison between the two treatment samples from either wild-type or SARM1-KO cells was done using Student’s *t*-test in B. ns, not significant; **, *p* < 0.01; ***, *p* < 0.001; ****, *p* < 0.0001.

**Figure 5 biology-13-00649-f005:**
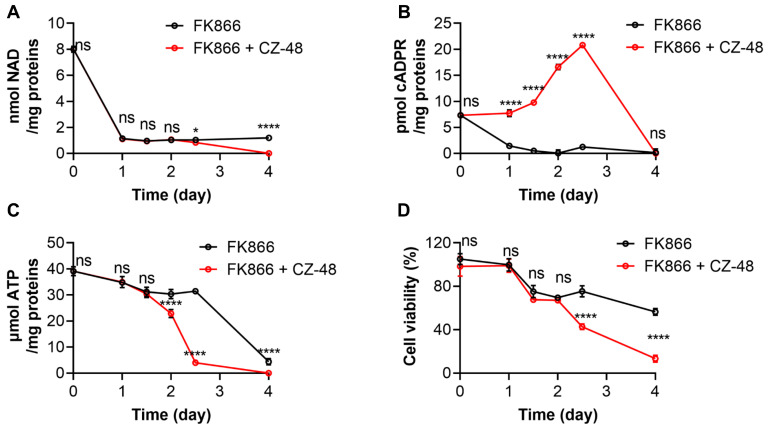
Kinetics of the treatment of CZ-48 and FK866 in U251 cells. (**A**–**D**) U251 cells were treated with 20 nM FK866 alone or in combination with 50 µM CZ-48 for a specified time period. NAD levels (**A**), cADPR levels (**B**), ATP levels (**C**), and cell viability (**D**) were measured by the cycling assay or SRB staining method. All experiments were independently performed three times, and data are presented as mean ± standard deviation. Student’s *t*-test, ns, not significant; *, *p* < 0.05; ****, *p* < 0.0001.

**Figure 6 biology-13-00649-f006:**
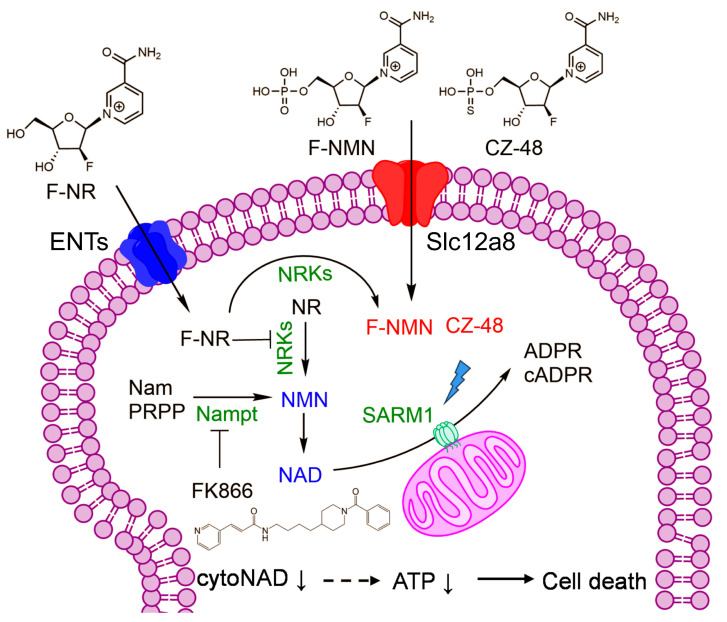
An illustrative model to explain how the fluorinated NAD precursors enhance FK866 cytotoxicity by activating SARM1 in glioblastoma cells.

## Data Availability

The metabolomics dataset and data analyzed during the current study are available from the corresponding author upon reasonable request.

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
