# Peer review of "The Fluorinated NAD Precursors Enhance FK866 Cytotoxicity by Activating SARM1 in Glioblastoma Cells"

_biology, 2024, doi:10.3390/biology13090649_

Round 1

Reviewer 1 Report

Comments and Suggestions for Authors

In this paper, He et al. investigated the effects of fluorinated versions of NAD precursors, F-NR on NAD depletion and the enhancement of F-NR on FK866 mediated cytotoxicity. The finding is interesting and useful in the field. However, the evidence for mechanism is confusing since all the data provided to illustrate the mechanism in this manuscript is related to CZ-48 not F-NR. The administration of F-NR lacks consistent criteria in term of concentration and duration of the treatment. There are some major and minor concerns before it is published in biology.

Major concerns,

1.        In Figure1, the authors investigated that F-NR competitively inhibits NRK1/2 which reduced NMN and NAD synthesis. However, in Figure 4, they illustrate that activation of SARM1 by F-NMF is actually important for NAD depletion.  They should address if the inhibition of NRK1/1 is biologically functional. Based on the figure 1G and figure 2A, even high concentrations of F-NR didn’t change NAD a lot although it is statistically significant.

2.        In Figure F-H, the concentration of F-NR used was 250 uM, in Figure 2, it was 100 uM, in Figure 3, they used 50uM. They should address why they used different concentration of F-NR in different experiments and how the concentrations were determined. And also why in some experiments the treatment was 12 hour, and in other figure it was 2 days or 4 days.

3.        In figure 4, they should test if depletion of SARM1 can cancel the impact of F-NMN on NAD consumption not CZ-48.

4.        In figure 5, the data is irrelevant to the topic.

Minor concerns,

1.        In figure 1E, ctrl should be included.

2.        Figure 4 A and B, statistical analysis is needed.

Author Response

Comments 1: In Figure1, the authors investigated that F-NR competitively inhibits NRK1/2 which reduced NMN and NAD synthesis. However, in Figure 4, they illustrate that activation of SARM1 by F-NMN is actually important for NAD depletion.  They should address if the inhibition of NRK1/1 is biologically functional. Based on the figure 1G and figure 2A, even high concentrations of F-NR didn’t change NAD a lot although it is statistically significant.

Response 1: We agree that the observed changes in NAD levels upon F-NR treatment might appear subtle, but they are indeed statistically significant. This observation is consistent with the understanding that the NAD salvage pathway primarily operates through the enzyme Nampt, while the pathway involving NRK1/2 is secondary and contributes less to the overall NAD pool. We believe that the current results were sufficient to support the conclusion that the decrease of NAD is the combined result of NRK inhibition and SARM1 activation.

Comments 2: In Figure 1F-H, the concentration of F-NR used was 250 uM, in Figure 2, it was 100 uM, in Figure 3, they used 50uM. They should address why they used different concentration of F-NR in different experiments and how the concentrations were determined. And also why in some experiments the treatment was 12 hour, and in other figure it was 2 days or 4 days.

Response 2: In short, the choice of concentrations and treatment durations was guided by preliminary data and the specific aims of each experiment. In detail, the concentration of 250 μM F-NR used in Figure 1F-H was based on preliminary dose-response studies that showed robust effects on NMN and NAD synthesis without causing significant cytotoxicity at a short time point (12 hours). For Figure 2, a lower concentration of 100 μM F-NR was used to investigate the effects on NAD levels over a longer period (2 days), aiming to assess the cumulative impact of F-NR on NAD metabolism. In Figure 3, U251 cells were treated with a series of concentrations of F-NR (including 50 μM) and monitored for cell viability over several days to determine the cytotoxic threshold and to assess the long-term effects on cell survival. The duration of treatment varied across experiments to address specific research questions: short-term treatments (12 hours) were used to capture the immediate effects of F-NR on NMN and NAD synthesis, while longer treatments (2 days or 4 days) were performed to investigate the sustained effects on NAD levels and cell viability.

We have included more description of these considerations in the Methods section and main text of the manuscript to provide clarity on our experimental design (Page 8, Methods section 2.9 and Page 11, Results section 3.1 and 3.2 line 1-2 in the file “manuscript_marked-up”).

Comments 3: In figure 4, they should test if depletion of SARM1 can cancel the impact of F-NMN on NAD consumption not CZ-48.

Response 3: We appreciate the reviewer's suggestion. Unfortunately, the U251 cells with SARM1 knockout and the compound F-NMN were lost during the recent lab move. Reconstructing the cells and resynthesizing the compound would require substantial effort and time. Given that both CZ-48 and F-NMN can activate SARM1 (Zhao et al., iScience, 2019), we believe that the results obtained with CZ-48 (Fig. 4D) partially address the question. These results demonstrate the importance of SARM1 activation in mediating NAD depletion.

Comments 4: In figure 5, the data is irrelevant to the topic.

Response 4: We understand the reviewer's concern. The kinetic measurements of various parameters in Fig. 5 provide crucial mechanistic insights into the cell death induced by CZ-48 and FK866. Therefore, we believe that the data in Fig. 5 are highly relevant to the overall topic of the manuscript.

Minor points:

Comments 5: In figure 1E, ctrl should be included.

Response 5: The control condition was indeed included and represented by the point at 100%, which corresponds to untreated enzyme. This baseline value serves as the reference point for all subsequent comparisons.

Comments 6: Figure 4 A and B, statistical analysis is needed.

Response 6: We have now performed the appropriate statistical tests and included the results in the updated figures. The statistical analysis confirms the significance of the observed differences between the groups. We have also added a description of the statistical methods used in the Methods section of the manuscript (Page 9, Method section 2.11; Page 14 and Page 16, figure legend in the file “manuscript_marked-up”).

Reviewer 2 Report

Comments and Suggestions for Authors

The present research introduces a novel approach using fluorinated versions of NAD precursors, specifically a compound named F-NR, which when combined with FK866, significantly boosts its effectiveness against glioblastoma cells. F-NR works by competing the endogenous metabolism of NR, leading to reduced NAD levels and enhancing FK866's ability to kill cancer cells. Another key finding is the role of SARM1, activated by one of F-NR's metabolite, which contributes to the enhanced cell-killing effect. The sequence of events—NAD depletion followed by energy loss and ultimately, widespread cell death—highlights the potential of this strategy to improve FK866's therapeutic impact. This research not only deepens our understanding of NAD metabolism in glioblastoma but also provides a potential strategy for treating the brain cancer.

The manuscript is precisely executed and nicely presented. However, some minor clarification needs to be provided before publication.

Figure 1 C, the two protein bands needs to be labled.

Figure 3A. The relative expression were mentioned. kindly explain which one is the reference sample which was used for normalization? Kindly re-check the normalization as if it is 2 delta delta (ΔΔCT) method relative quantification

Author Response

Comments 1: Figure 1 C, the two protein bands needs to be labled.

Response 1: We appreciate the reviewer's feedback. The labels for the respective proteins were indeed placed above the gel images. To ensure clarity and ease of interpretation, we have now added clear labels directly adjacent to the corresponding bands within the figure.

Comments 2: Figure 3A. The relative expression were mentioned. kindly explain which one is the reference sample which was used for normalization? Kindly re-check the normalization as if it is 2 delta delta (ΔΔCT) method relative quantification

Response 2: We appreciate the reviewer's feedback. We apologize for the confusion. Indeed, we did not use the ΔΔCT method for relative quantification. Instead, we employed the ΔCT method. This means that we normalized the expression of the genes of interest to a housekeeping gene within the same sample, rather than to a reference sample. The ΔCT values were then used to calculate the relative expression levels. We have now clarified this methodology in the Methods section of the manuscript (Page 7, the last line in the file “manuscript_marked-up”).

Round 2

Reviewer 1 Report

Comments and Suggestions for Authors

The authors have addressed all my concerns.